# Stability and Feasibility of Dried Blood Spots for Hepatitis E Virus Serology in a Rural Setting

**DOI:** 10.3390/v14112525

**Published:** 2022-11-15

**Authors:** Joakim Øverbø, Asma Aziz, K. Zaman, Cathinka Halle Julin, Firdausi Qadri, Kathrine Stene-Johansen, Rajib Biswas, Shaumik Islam, Taufiqur Rahman Bhuiyan, Warda Haque, Synne Sandbu, Jennifer L Dembinski, Susanne Dudman

**Affiliations:** 1Norwegian Institute of Public Health, NO-0213 Oslo, Norway; 2Department of Microbiology, Institute of Clinical Medicine, University of Oslo, NO-0424 Oslo, Norway; 3International Centre for Diarrheal Diseases Research, Dhaka 1212, Bangladesh; 4International Vaccine Institute, Seoul 08826, Republic of Korea; 5Oslo University Hospital, NO-0424 Oslo, Norway

**Keywords:** dried blood spot, DBS, hepatitis E, HEV, vaccine, serology, IgG

## Abstract

Hepatitis E virus (HEV) is the most common cause of acute viral hepatitis worldwide. In many low-income countries it causes large outbreaks and disproportionally affects pregnant women and their offspring. Surveillance studies to find effective preventive interventions are needed but are hampered by the lack of funding and infrastructure. Dried blood spots (DBS) offer an easier and more robust way to collect, transport, and store blood samples compared to plasma/serum samples, and could ease some of the barriers for such studies. In this study we optimize an HEV IgG ELISA for DBS samples and validate it on 300 paired DBS and plasma samples collected in rural areas of Bangladesh from participants in a HEV vaccine study. We demonstrate that HEV IgG in blood stored as DBS is stable for two months at up to 40 °C, and for five freeze-thaw cycles. The specificity was 97% and the overall sensitivity of the DBS assay was 81%. The sensitivity was higher in samples from vaccinated participants (100%) compared to previously infected participants (59%), reflecting a positive correlation between IgG titer and sensitivity. We found a strong correlation between DBS and plasma samples with an r2 of 0.90, but with a higher degree of difference between individual paired samples. Our study shows that DBS offers a stable alternative to plasma/serum for HEV IgG measurements and can facilitate serological studies, particularly in resource limited areas.

## 1. Introduction

Hepatitis E virus (HEV) is a major cause of acute hepatitis around the world, but with a disproportionately high burden in low-income countries, where pregnant women and their offspring are at particular risk of severe disease and death due to HEV genotype 1 in particular [1]. This genotype is common in many low-income countries such as Bangladesh [2], while genotypes 3 and 4 dominate in high- and middle-income countries [1]. Several public health interventions, including a new vaccine [3], could probably alleviate some of the burden. However, large studies of HEV transmission and immunity are needed to establish effective interventions [4]. 

Serosurveillance studies are important for understanding the effect of infection control measures, where factors like incidence, prevalence, and herd immune levels may be investigated [5]. Such studies usually depend on plasma or serum samples collected by venipuncture, requiring trained health personnel, cold chain for transportation, and storage for samples. These factors are likely to limit the number and sizes of serological studies, particularly in low-income countries. 

An alternative, easy, and cost-effective sample type is dried blood spots (DBS), where capillary blood from a fingertip punction is collected on filter paper and dried. The method is minimally invasive and can be easily performed by anyone, even the donor. The cards require little space and can be stored at room temperature for longer periods [6]. DBS have been used successfully for measuring antibodies against other viruses such as HIV, Hepatitis B and C, and SARS-CoV-2, and is well suited for field studies in low-income countries [7]. However, the accuracy of measuring serological response to infection or vaccination from DBS samples seem to vary between different viruses [8], and hence need to be evaluated for the virus in question. A previous small study provided promising results in detecting HEV-IgG from DBS samples of acute HEV cases [9]. However, using DBS for IgG detection in people with previous HEV infection or vaccination has not been studied. 

The main objective of this study was to optimize and validate DBS as sampling material, in a low-income country setting, to detect HEV IgG antibodies for use in serosurveys and vaccine trials. The validation included testing the effect of freeze-thawing of DBS, as well as long time storage at room temperature at 20 °C (normal) and 40 °C (tropical climate).

## 2. Materials and Methods

This study is part of a large randomized controlled trial evaluating the effectiveness of a hepatitis E vaccine (HEV 239) in Bangladeshi women [9]. The vaccine is based on a 239 amino acid long recombinant HEV genotype 1 peptide, which encodes the capsid protein [10]. The clinical trial is conducted by the Norwegian Institute of Public Health (NIPH) (Oslo, Norway) and the International Centre for Diarrheal Disease Research, Bangladesh (icddr,b).

### 2.1. Sample Material

Three sources of sample material were used (Figure 1):

Group 1: Whole blood samples were drawn by venipuncture into EDTA-coated tubes from 10 confirmed anti-HEV negative volunteers. The 10 whole blood samples were split into separate tubes and spiked with WHO reference reagent for HEV IgG antibody (NIBSC code: 95/584) to achieve the relevant anti-HEV IgG levels for further analysis. 

Group 2: Whole blood samples were drawn from May to July 2017 from 100 healthy participants (50 men and 50 women) aged 16 to 39 from two villages (Sepaikandi and Naburkandi) in Matlab, Bangladesh, participating in a pilot study of the HEV 239 vaccine trial. The participants were vaccinated with HEV 239 or a control vaccine (Hepatitis B) as a 2-dose regimen at day 0 and 30 (described in ClinicalTrials.gov Identifier: NCT02759991 (pilot-study)). Blood samples were drawn before vaccination and 30 days after the second vaccine dose by venipuncture into EDTA-coated tubes. The samples were immediately refrigerated and kept at a temperature between 2 °C and 8 °C during transport to the laboratory (max 24 h transport time). 

Group 3: Blood samples were drawn from 50 healthy women aged 16 to 39 randomly selected from a group of 19,640 women from 67 villages in Matlab, Bangladesh, participating in the main HEV 239 vaccine trial [11]. The women received either HEV 239 or a control vaccine (HBV) at day 0, 30 and 180. Paired venous and capillary blood were collected from each participant before vaccination and 30 days after the third vaccine dose. Venous blood was drawn by venipuncture into EDTA-coated tubes. Capillary blood was collected by finger prick onto a Whatman 903 filter paper card (Sigma-Aldrich, St. Louis, MO, USA).

### 2.2. Specimen Preparation

DBS samples were prepared from venous whole blood (groups 1 and 2) or capillary blood directly applied from fingertip puncture (group 3). The whole blood samples were transported to the laboratory at a temperature between 2 °C and 8 °C and prepared within 24 h after venipuncture. Approximately 50 µL of whole venous or capillary blood was transferred to each circle on the Whatman 903 filter paper card (Sigma-Aldrich). The blood was allowed to air-dry flat, for at least half an hour, and then in a DBS drying rack until dry (at least four hours), then placed in individual zippered storage bags with two silica desiccant packs and a humidity indicator. Plasma was isolated from the remaining venous blood by centrifugation. Both DBS bags and plasma samples were stored at −80 °C until analysis.

### 2.3. DBS Extraction

Circles of 3.2 mm in diameter (equivalent to approximately 1.5 µL plasma [12]) were punched out from one spot on a DBS card and diluted in Wantai ELISA sample diluent to obtain the desired dilution (for 1:40, 3 punches in 150 µL diluent was used). The samples were incubated at 4 °C overnight before being centrifuged at 698 g (2500 RPM) for 7 min the next day. The supernatant was transferred to a new tube and analyzed within 24 h (stored at 4 °C) or stored at −80 °C until analysis. 

### 2.4. Serology

DBS eluates and plasma samples were tested for anti-HEV IgG using Wantai HEV IgG ELISA (Beijing Wantai, Beijing, China). This enzyme-linked immunosorbent assay uses a peptide (PE2) encoded by a structural region of ORF-2 derived from a Chinese isolate of HEV genotype 1 [13] and is validated for detecting IgG-class antibodies to hepatitis E virus in human serum or plasma. The original dilution is 10 µL plasma/serum added to 100 µL diluent. The optical density (OD) values are reported between 0–3.5, and the original cut-off value (CO) is calculated by adding 0.16 to the average OD of the negative internal controls according to the manufacturer’s instructions. 

The initial dilution step in the ELISA protocol was optimized for DBS as described above, while the remaining procedure for DBS eluates and the complete procedure for plasma samples was done according to the manufacturer’s instructions. If the results were above the limit of quantification of the kit (OD > 3.5) the samples were rerun at higher dilutions (20, 160, 400, and 1000). 

To quantify all IgG results in WHO international units (WU/mL), we made separate standard curves for plasma and DBS samples using spiked samples and used a 5 parametric logistic regression described by Gottschalk and Dunn to standardize the results [14]. 

### 2.5. Optimization (Dilution and Cut-Off) 

The standard dilution for Wantai HEV IgG ELISA using serum/plasma is 1:11. For DBS, dilutions between 1:11 and 1:200 (including the extraction process) were evaluated. The optimal dilution was determined by maximizing the difference in OD values between negative and weak positive samples, and the ease of processing.

From the results of paired samples from group 2, we evaluated three methods for calculating the optimal cut-off value above which DBS samples were considered positive: (1) the assay protocol by dividing the optical density value by the cut-off value (OD/CO) and values above one counted as positive, (2) two standard deviations (SD) above the mean DBS OD/CO of the negative samples, and (3) a Receiver Operating Curve (ROC) analysis by evaluating a range of possible cut-off values and their corresponding sensitivity and specificity.

### 2.6. Precision, Linearity and Stability (Group 1)

Inter and intra-laboratory coefficients of variation (%CV) were determined by running spiked samples (0.2–30.0 WU/mL) on nine ELISA plates in two different laboratories (NIPH and icddr,b). Precision limits of ≤25% were deemed acceptable.

Two DBS samples with 30 WU/mL (spiked) HEV-IgG were diluted 1.5, 3, 6, 12, 24, and 48 times, and the percentage recovery relative to the theoretical values linearity were calculated. 

DBS samples from spiked whole blood with levels of 2.5, 10, and 20 WU/mL were stored in separated zip bags with desiccant packs at temperatures −80, −20, 20, and 40 °C for up to 60 days (6 months for −20 °C and −80 °C). A separate set of samples with the same values underwent up to five freeze-thaw cycles. Each cycle consisted of a thaw period of 2 h at room temperature and at least 12 h subsequent storage at −80 °C. All paired samples were analyzed together, and HEV-IgG levels were compared to the matched samples stored at −80 °C. 

### 2.7. Clinical Performance (Groups 2 and 3)

Qualitative results from paired DBS and plasma samples from groups 2 and 3 were used to calculate the sensitivity and specificity using the optimized dilution and cutoff values. 

The correlation between antibody concentration in DBS and plasma was calculated by Lin’s concordance correlation coefficient as recommended by Watson and Petrie [15], using the “concord” module in Stata [16], and by linear regression analysis. 

All statistical analyses were performed in STATA [17]. The Pearson chi-squared test was used to check for significant differences between groups (binomial results). 

## 3. Results

Seronegative blood from 10 participants (group 1) was used for preclinical validation and optimization. For the clinical validation, 300 paired DBS and plasma samples from 150 participants were analyzed (groups 2 and 3). In group 2, the seroprevalence was 0.33 (95% CI: 0.24–0.43) and the geometric mean IgG titer (GMT) in positive samples before vaccination was 3.33 (2.05–5.40) WU/mL. In group 3, the seroprevalence was 0.38 (0.25–0.53) and the GMT in positive samples before vaccination was 0.74 (0.41–1.34). 

### 3.1. Optimization of ELISA Assay for DBS

By analyzing serial dilutions of spiked venous samples, we determined the laboratory specific preferred dilution to be 1:40 and 1:60 at our two laboratories. OD/CO results from paired plasma and DBS from group 2 were used to establish the DBS cut-off value. The area under the curve in the ROC analysis was 0.95 (0.90–0.97) (Figure 2) and a cut-off value of 1.6 (OD/CO) corresponding to 0.6 WU/mL (2 SD above mean value from negative samples), was deemed favorable for our purpose. The lower limit of quantification was 0.6 WU/mL in DBS and 0.1 WU/mL in plasma. The upper limit of quantification was 30 WU/mL for DBS and 8 WU/mL for plasma. 

### 3.2. Precision, Linearity and Stability

Inter-laboratory precision was assessed using spiked samples (group 1) with IgG levels between 0.5 and 30 WU/mL in two parallels for 9 ELISA runs. The overall coefficient of variation (%CV) was 24%, with a higher %CV in weak positives (0.5–2) of 41%, and lower in strong positives (2–30) of 6.3%. In the same way, the intra-laboratory variability within our two laboratories was found to be 19.2%.

Serial dilutions of a high positive sample (30 WU/mL) yielded a fairly accurate linearity (Table 1).

For the stability testing, storage for up to 60 days at 20 °C and 40 °C did not result in any major (%CV above 10%) changes in OD/CO, except for the low sample at 40 °C with a %CV of 18.5% after 30 days and 20.4% after 60 days (Figure 3a,b). Further, for freeze-thawing stability testing we did not detect any sign of degradation of the analyte in DBS samples for up to five freeze-thaw rounds (Figure 3c). The mean %CV was 2.26 after five cycles and the highest %CV was 10.1, occurring in the sample with low antibody levels after two cycles. Storage for 6 months at −20 °C did not result in any substantial alterations in OD/CO values compared to storage at −80 °C (%CV 2.1%).

### 3.3. Clinical Performance

The overall sensitivity of DBS compared to plasma in groups 2 and 3 combined was 81% (95% CI, Clopper-Pearson, 73.8–87.0) and specificity 97% (91.3–99.0). The sensitivity was lower for detection of past infection compared to recent vaccination (Table 2 and Table 3) (*p* < 0.01). The difference in sensitivity of unvaccinated participants in groups 2 and 3 (68% vs. 55%) was not significant (*p* = 0.25). Comparing DBS to plasma, used as the gold standard, we detected 5 false positive and 29 false negative DBS results in a total of 300 samples (Table 2).

To evaluate the clinical significance of the reduced limit of detection, we calculated separate sensitivities for different IgG levels. The results were compared to two different IgG thresholds (WU/mL) of which: (1) 90% of people infected in the past two years will have IgG levels above, (2) 90% of people vaccinated two years prior will have IgG levels above (Figure 4). The thresholds were estimated on results from the HEV vaccine pilot trial (group 2) (submitted for publication).

The seroprevalence estimated by plasma samples was 0.36 (95% CI, Clopper Pearson, 0.28–0.44) and 0.25 (0.18–0.33) for DBS (*p* = 0.04).

A linear regression of DBS and plasma produced a coefficient of determination r2 of 0.90 (Figure 5). Lin’s concordance correlation coefficient was 0.94 (95% CI 0.92–0.95), with similar results in vaccinated (0.92) and unvaccinated (0.94) participants. The average difference was 6.3 WU/mL higher for plasma samples, with a 95% limit of agreement (Bland and Altman) between −70.7 and 58.1. This indicates a good correlation of WU/mL between the two sampling methods, but with a high degree of difference between individual paired samples. The geometric mean of IgG levels for plasma was 14.95 WU/mL (95% CI 10.96–20.36), and for DBS 14.75 WU/mL (95% CI: 10.70–20.35).

## 4. Discussion

This study shows that DBS from capillary or venous blood is a reliable sampling alternative to plasma for HEV IgG measurement and can be used to detect vaccine responses and past infections for at least up to two years. We found DBS to be stable for two months at temperatures up to 40 °C and after five freeze-thaw cycles. Our study demonstrates that DBS can be successfully used to collect and transport blood samples in rural Bangladesh. A single fingerprick reduces the burden for blood donors and the demand for trained health personnel. The stability of IgG levels and small sample sizes reduces the cost and logistic hurdles of cold chain transport and storage [7]. 

Our results show that HEV serosurveys and vaccine response studies can be added to the extensive and growing list of applications of DBS [6]. Although we obtained a lower sensitivity (81%) compared to the mean sensitivity, in a recent review of similar studies (99%) [7], our specificity (97%) was higher compared to the same study (95%). The reduced sensitivity to detect low levels of IgG should be considered if this method is used for serosurveys, as DBS resulted in a significant reduction in estimated seroprevalence compared to results from the plasma samples. Still, it is likely to provide useful data globally, if the results are carefully interpreted and adjusted, especially at a population level.

The observed increase in optical density from IgG negative samples is likely a matrix effect caused by hematocrit or substances in the filter paper [18]. Such effects might be a more prominent issue when using ELISA assays that originally require a low dilution of plasma, such as ours. In order to avoid many false positive samples, we increased the dilution and cut-off value for DBS compared to plasma. This resulted in a higher limit of detection and a higher upper limit of quantification, thereby increasing the overall quantifiable measurement range of the assay. 

Our study included DBS samples from both venous blood (group 2) and capillary blood (group 3). The lower (non-significant) sensitivity observed in group 3 could be a result of this difference, but the 4.5 times difference in GMT in positive samples prior to vaccination between the two groups is a more likely cause of this difference. 

The inter-laboratory %CV of ELISA results from DBS reported in the literature varies from below 10 [19] to above 40 [20]. The measured %CV of 24 in our study is relatively high for a quantitative method, but within expectations of a semi-quantitative method due to the sampling methodology used. We suggest that individual IgG level results from DBS should be interpreted with caution as laboratory precision for DBS samples was at the lower end of our defined limits (25% CV) and the relatively high degree of variance we observed between individual paired DBS and plasma samples. However, a high correlation and similar geometric means between DBS and plasma should allow for a high confidence in interpreting quantified IgG levels on group levels.

A limitation of our study is the lack of hematocrit measurements in DBS samples, which could have helped us explore its impact on the specificity, both between individual samples and on the mean values observed. It is possible that the specificity of the described ELISA method can be different in other populations with different average hematocrit levels. Another limitation of the study is that the stability testing was only conducted on one sample for every timepoint, and only lasted for two months. However, we see a trend showing that the DBS is stable throughout the 2-month period tested, but may decrease slightly over time in high temperatures, as expected. 

Our validation study is limited to one type of filter paper and one diluent without evaluating alternatives. Other filter papers or diluents may provide better results and could be explored in future studies. 

The procedure described in our paper can be considered a semi-quantitative method and is likely to produce reliable data on mean antibody titers in groups and help determine the prevalence of infections or vaccine responses within at least two years after the event. 

## 5. Conclusions 

This study provides the first evidence that vaccine and infection-induced HEV IgG can be accurately detected from blood collected as DBS within a 2-year period after the event. DBS also proved to be a stable storage media for HEV IgG, withstanding temperature variation and longtime storage. Based on these findings, this DBS method was implemented in a large HEV vaccine trial currently ongoing in Bangladesh. DBS offers a robust and effective sample method for monitoring HEV prevalence, population immunity, and effectiveness of interventions such as vaccination. The observed decrease in sensitivity of weak positive DBS samples should be accounted for in serosurveys. By using DBS rather than plasma/serum, screening for HEV IgG would become more feasible and accessible, thus facilitating much needed research and surveillance programs for HEV.

## Figures and Tables

**Figure 1 viruses-14-02525-f001:**
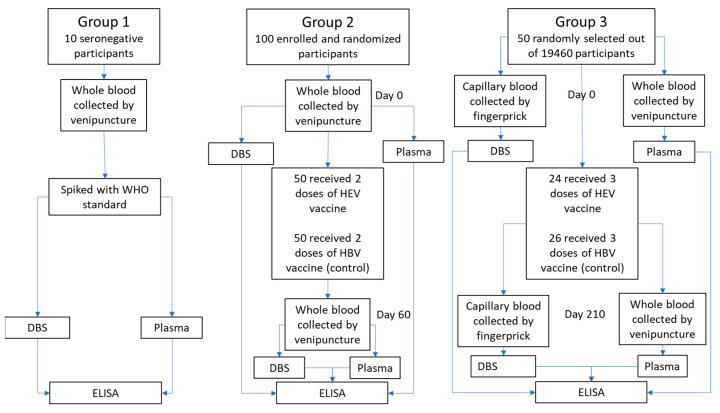
Flow chart of sample processing for the three different source materials used in this study.

**Figure 2 viruses-14-02525-f002:**
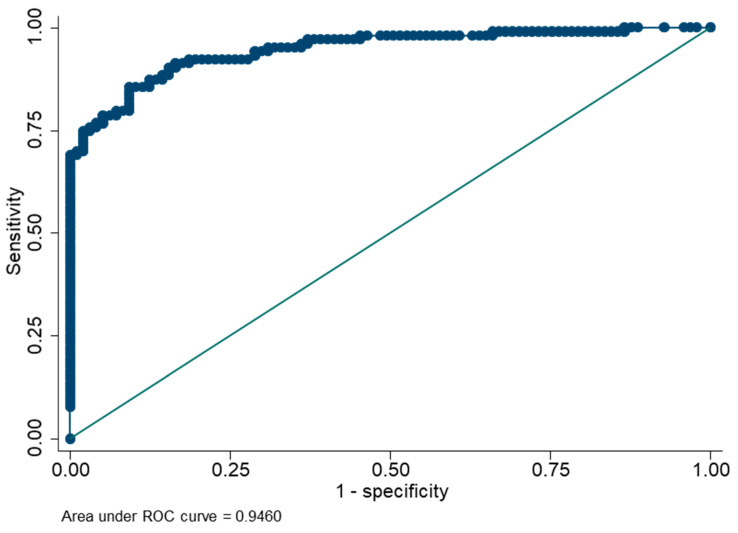
ROC analysis for ELISA results (OD/CO) from DBS compared to plasma, based on results from 200 paired DBS and plasma samples.

**Figure 3 viruses-14-02525-f003:**
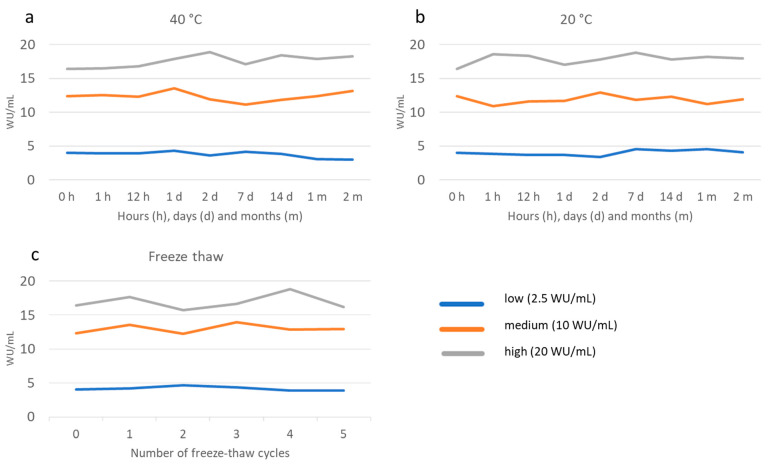
Stability of DBS stored at 40 °C (**a**) and 20 °C (**b**) for up to 60 days and a various number of freeze-thaw cycles (**c**) as measured by HEV IgG in international units (WU/mL) at various time points.

**Figure 4 viruses-14-02525-f004:**
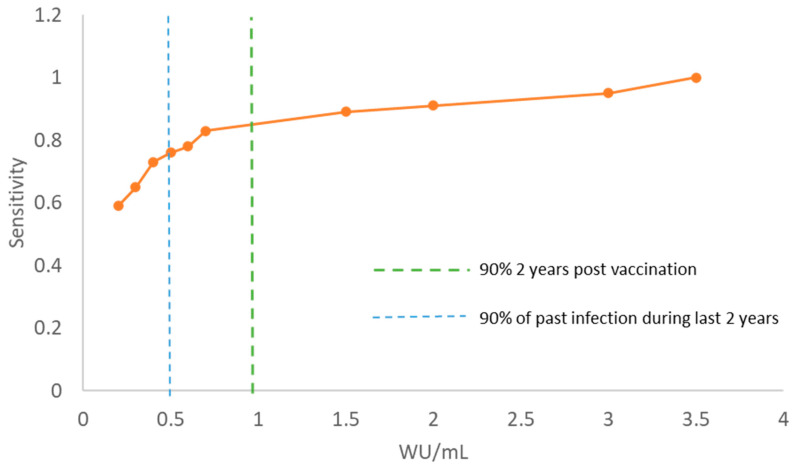
Positive correlation between IgG titer (WU/mL) and the sensitivity from DBS sample (orange line). Dotted lines represent the level of IgG of which 90% of people with an infection (blue) or vaccination (green) will be above.

**Figure 5 viruses-14-02525-f005:**
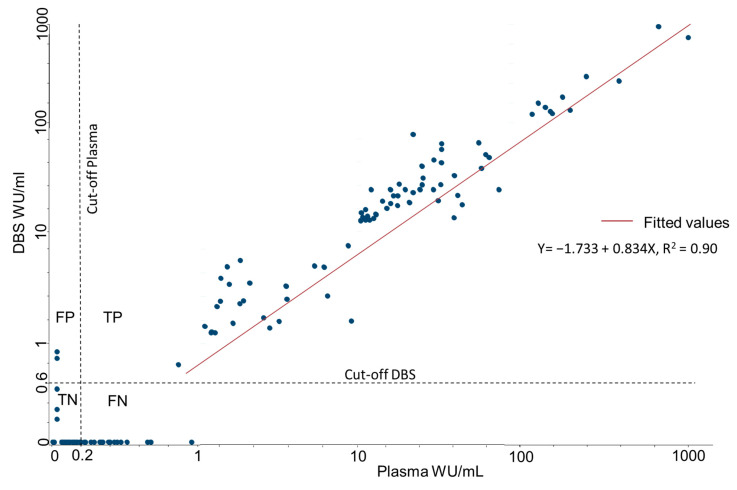
Correlation between IgG concentrations obtained from plasma and DBS, standardized to WU/mL, based on the results from 300 paired DBS and plasma samples. FP = False positive, TP = True positive, TN = True negative, FN = False.

**Table 1 viruses-14-02525-t001:** Dilution linearity of DBS samples starting with 30 WU/mL HEV IgG analyzed in parallel (DBS 1 and 2). Results are presented in IgG level (WU/mL) and the difference between the observed and expected IgG level in percent (% expected value).

DBS		Dilution Factor				
		Undiluted	1.50	3	24	48
1	WU/mL	30.7	19.4	10.6	1.0	0.6
	% Expected value	102	97	106	79	93
2	WU/mL	27.3	21.5	9.6	1.6	0.9
	% Expected value	91	107	96	125	137
Mean	% Expected value	97	102	101	102	115

**Table 2 viruses-14-02525-t002:** Qualitative results from paired plasma samples (used as gold standard) and DBS in vaccinated and unvaccinated participants. (group 2 = 200 samples, group 3 = 100 samples).

	Plasma					
	Group 2		Group 3		Group 2 + 3	
DBS	Positive	Negative	Positive	Negative	Positive	Negative
Total						
Positive	84	3	41	2	125	5
Negative	16	97	13	44	29	141
Unvaccinated						
Positive	34	3	16	2	50	5
Negative	16	97	13	44	29	141
Vaccinated						
Positive	50	0	25	0	75	0
Negative	0	0	0	0	0	0

**Table 3 viruses-14-02525-t003:** Clinical performance of DBS vs. plasma in detecting positive HEV IgG samples in HEV 239 vaccinated (Vac) and unvaccinated (Unvac) participants.

	Group 2			Group 3			Group 2 + 3		
	Vac	Unvac	Total	Vac	Unvac	Total	Vac	Unvac	Total
Sensitivity (%)	100	68	84	100	55	76	100	63	81
Specificity (%)	100	97	97	100	96	96	100	97	97
PPV	1.00	0.92	0.97	1.00	0.89	0.95	1.00	0.91	0.96
NPV	1.00	0.86	0.86	1.00	0.77	0.77	1.00	0.83	0.83

## Data Availability

Not applicable.

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
