# Peer review of "Stability and Feasibility of Dried Blood Spots for Hepatitis E Virus Serology in a Rural Setting"

_viruses, 2022, doi:10.3390/v14112525_

Round 1

Reviewer 1 Report

Overall, the authors have designed a correct evaluation of DBS for HEV serology testing using 3 different groups. They evaluated this innovative sample from naturally infected subjects and participants of a HEV vaccine trial in Bangladesh and describe the evaluation process in detail. In addition, prevalence of natural HEV infection is studied but no information or discussion is given about these results.

The use of English, particularly in the discussion and conclusions, should be reviewed.

Specific comments:

Introduction

Introduce please the issue of genotypes commonly found in Bangladesh, where the samples were collected, and the genotype formulation of the vaccine evaluated.

Line 44, add after countries “and people without contact with the healthcare environment in high-income countries”

Line 46, replace “easier” with “easy”

Line 50, replace “DBS has been used” by “DBS have been used”

Lines 55-56, replace “Still, use of DBS in IgG detection” by “However, use of DBS for IgG detection”

Lines 59-60, replace “The validation including” by “The validation included”

Methods

Please confirm if the acronym “icddr,b” is correct without capital letters.

Figure 1. Replace “recieved” by “received” 4 times (Groups 2 and 3)

2.3 DBS extraction. Please, indicate the volume used of Wantai ELISA sample diluent used and the time of storage at -80ºC (up to…).

2.4 Serology. Indicate the volume of eluate and plasma samples used for the tests.

2.5 Optimization

Line 134. Replace “was 1:11” by “using serum/plasma is 1:11”

Ines 134-135. Explain if DBS dilutions (1:11 – 1:200) include that of DBS extraction process or not.

Line 158. 2.7. Clinical performance. Add “(groups 2 and 3)” as in 2.6

When statistical methods are explained, did you perform a normality test?

Results

Line 176. Change “2SD” by “2 SD”

Line 184. Delete “for”

Lines 185-186. Describe major mistakes occurred (positive samples by plasma testing not detected in DBS samples (false negative), as well, as false positive in DBS samples).

Line 187. Replace “between” by “within” if intra-laboratory data is shown.

Table 2. Add in the table, in the text or preferably build a new table that displays median and interquartile range of HEV IgG levels (WU/mL) for plasma samples within groups 2, 3 and 2+3 for each category (positive, negative –total, unvaccinated and vaccinated-). Carry out statistical analysis if possible

Table 2. Review total numbers of Group 2 as there might be some mistake.

Table 3. Replace “specifisity” by “specificity”

Analyze if there is significant difference between the sensitivity of unvaccinated of groups 2 vs 3 (0.63 vs 0.54)

Consider including results about prevalence of HEV infection found using plasma and DBS in Bangladesh 

Discussion

Line 244. Delete “60d”

Line 249. Replace “hurdles of transporting, collection and storing the samples” by “hurdles of sample transportation, collection, and storage”

Line 251. Replace “for DBS” by “of DBS”

Line 256. Replace “marked” by “market” and remove repetition of “it is likely to”.

Discuss about the advantages of DBS use also in high-income countries

Line 267. Remove repetition of % symbol 

Lines 269-270. Please, reformulate this sentence, it´s not well understood

Discuss about the %CV or the feasibility of HEV IgG detection and other methods developed for DBS work (HIV, hepatitis or SARS Ab detection).

Address the issue of the prevalence found in this area, the circulating genotypes and influence in the method of detection, and compare the prevalence data with previous reports from this area.

Add as a limitation or discuss the different procedures used for DBS preparation for groups 2 and 3, and how this issue may influence the observed sensitivity results.

Explain if negative samples were evaluated to check specificity as DBS material or the diluent used may have some influence. Otherwise, consider to include this as a limitation.

Include additional references for the proposed new discussion topics and for an explanation of the limitations of the study.

Conclusions

Reformulate this paragraph to highlight your main findings and review the use of English.

Reviewer 2 Report

The study by Øverbø describes the application of dried blood spots for performing HEV IgG serology using the Wantai anti-HEV test. Use of DBS samples has a clear advantage in several settings and the study convincingly shows that DBS samples can be reliably tested, albeit with a lower sensitivity (which should not be a problem for determining vaccination responses but may be more problematic for epidemiological studies). The study is well conducted, the manuscript well written. The limitations, including a lowering of the sensitivity, are well described. 

I only have a few comments.

- Line 110: the volume in which DBS samples were extracted is lacking. If 11x dilutions were used and one spot corresponds to 1.5 µl this volume might have been 16.5µl but that seems very low (and would probably be absorbed).  So perhaps a fixed higher volume was used and dilution was calculated relative to this initial dilution?

- Line 121: The cut-off for the Wantai test is 0.16 plus the average negative control (with a minimum of 0.03) which in practice is always 0.16+0.03=0.19 and not 0.03 plus the average negative control. If you used a different cutoff please explain why.

- Line 253-256: Although I agree that the Wantai test is among the most sensitive on the market this does not automatically correspond to reliable data in all cases. The seroprevalence may be strongly underestimated in a serosurvey in a population with a constant exposure to HEV. I agree with the observation that detection in DBS is reliable up to two years, but two years is still short, and IgG remains positive for much longer (probably decades) if plasma or serum are tested.  Please phrase this more carefully. 

- Line 256: you mean market, not marked.

Reviewer 3 Report

HEV is a significant cause of acute hepatitis globally. Monitoring the spread of this virus is essential. The authors optimised IgG HEV ELISA and adopted DBS as the proper blood collection method for seroepidemiological monitoring of HEV exposures, including tracking herd immunity and immune response against the HEV vaccine. A simplified blood sample collection method is appropriate for broad seroepidemiological studies in remote areas lacking resources. The research is described clearly, and the obtained results are reliable.

Lanes 22-23. The results of sensitivity and specificity are better expressed as percentages.

Lane 25. I suggest r2 be rewritten as r2.

Lanes 70-73. How many samples in this group were used?

Lanes 75-83. The description of group 2 is missing information about the collection of DBS. The information in the text does not match the information in Figure 1.

Lane 119. The HEV antigen used in the commercial test should be added to the text, but it is not necessary in this lane.

Lane 142. Explanation of how ROC analysis is used for cut-off value calculation.

Lane 149. “(linearity)” between words “values …. were” could be added.

Lane 180. the caption for Figure 2 must be below the image. The number of samples used in the figure must be given in the caption.

Lane 192. What means “(1 and 2)”?

Table 1. The table caption “% expected value” should be explained.

Lane 200. What do you mean by “degradation of DBS samples”? The degradation of analyte? This should be added to the text.

Lane 211. The sensitivity of what? Missing information about the test and type of samples used.

In table 3 the sensitivity and specificity are better expressed as %.

Lanes 238-239. The number of samples used in the figure must be given in the caption.

Throughout the manuscript, text proofreading mistakes have been noticed. Note the spelling of the spaces between the number and the various units.
